

# A quantitative approach to determine the taxonomic identity and ontogeny of the pycnodontiform fish *Pycnodus* (Neopterygii, Actinopterygii) from the Eocene of Bolca Lagerstätte, Italy

John Joseph Cawley[1], Giuseppe Marramà[1], Giorgio Carnevale[2] and Jürgen Kriwet[1]

[1] University of Vienna, Department of Palaeontology, Vienna, Austria
[2] Dipartimento di Scienze della Terra, Università degli Studi di Torino, Turin, Italy

Corresponding author
John Joseph Cawley,
john.cawley@univie.ac.at

## ABSTRACT

**Background:** The pycnodontiform fish *Pycnodus* is one of the representatives of the highly diverse actinopterygian fish fauna from the early Eocene Bolca Lagerstätte, representing one of the youngest and thus last occurrences of this extinct neopterygian clade. This genus has historically been used as a wastebasket taxon in regards to poorly known pycnodontiform fossils. Authors have argued over the specific status of the Bolca Lagerstätte *Pycnodus* in terms of how many species are contained within the genus with some arguing for multiple species and others suggesting lumping all Bolca specimens together into one species.
**Methods:** Here, we use a quantitative approach performing biometric and geometric morphometric analyses on 52 specimens of *Pycnodus* in order to determine if the morphological variability within the sample might be related to inter- or intraspecific variation.
**Results:** The analyses revealed that the variations of body shape, morphometric and meristic characters cannot be used to distinguish different morphotypes. On the contrary, our results show a remarkable link between shape and size, related to ontogeny.
**Discussion:** Differences in body shape of small (juvenile) and large (adult) individuals is probably related to different microhabitats occupation on the Bolca reef with juveniles sheltering within crevices on the reef and adults being more powerful swimmers that swim above the coral. Taxonomically, we suggest that the Bolca *Pycnodus* should be referred to strictly as *Pycnodus apodus* as this was the name given to the holotype. Additionally, an overview of species assigned to *Pycnodus* is given.

## INTRODUCTION

Pycnodontiform fishes were a highly successful group of neopterygian fishes that colonized shallow marine, brackish, and freshwater habitats from the Norian to the

middle Eocene during ca. 170 Ma (*Tintori, 1981*; *Longbottom, 1984*; *Poyato-Ariza et al., 1998*; *Kriwet, 2005*). They were particularly diverse during the Late Cretaceous when they showed the highest degree of morphological diversity (*Marramà et al., 2016a*; *Cawley & Kriwet, 2018*). Pycnodonts underwent a severe drop in their diversity and disparity at the end of the Cretaceous, and the last representatives survived in restricted biotopes until the Middle Eocene (*Poyato-Ariza, 2005*; *Marramà et al., 2016a*). One of the last Palaeogene representatives is *Pycnodus apodus* (*Volta, 1796*), which is represented by several complete and articulated skeletons from the early Eocene (late Ypresian, c. 49 Ma) (*Papazzoni et al., 2014*; *Marramà et al., 2016b*) Bolca Koservat-Lagerstätte. This deposit yielded a huge amount of exquisitely preserved fishes, which are housed today in several museums and research institutions around the world, and that are represented by more than 230 bony and cartilaginous fish species (see e.g., *Blot, 1987*; *Blot & Tyler, 1990*; *Bannikov, 2004*, *2006*, *2008*; *Bannikov & Carnevale, 2009*, *2010*, *2016*; *Carnevale & Pietsch, 2009*, *2010*, *2011*, *2012*; *Carnevale et al., 2014*, *2017*; *Marramà & Carnevale, 2015a*, *2015b*, *2016*, *2017*; *Marramà et al., 2017a*, *2017b*).

*Pycnodus apodus* has a long and complex taxonomic history (see e.g., *Blot, 1987*; *Poyato-Ariza & Wenz, 2002*). *Volta (1796)* originally designated it as *Coryphaena apoda*. *Blainville (1818)* subsequently redescribed the same specimens without illustrations, and erected for them the taxon *Zeus platessus*. Finally, *Agassiz (1833–1844)* created the genus *Pycnodus* for these specimens but kept the specific name of *Blainville (1818)*. *Agassiz (1833–1844)* noted that the existence of small specimens with a swelling of the forehead to be juveniles of *P. platessus*. *Heckel (1856)* erected using the same material as Agassiz (but probably also including other specimens) from Bolca a second species of *Pycnodus*, *P. gibbus*, due to differential characters such as the presence of a gibbosity on the forehead, higher vertebrae length to body depth ratio than *P. platessus* and the body depth being one and a half times that of the body length in contrast to *P. platessus* having a body depth half that of the length. Another character not explicitly mentioned in the text but was drawn (*Heckel, 1856*, Plate 8, Fig. 4) is that *P. gibbus* has two interdigitations between the vertebrae while *P. platessus* has three to four. More recently, *Blot (1987)* examined specimens that were labeled *P. platessus* in various institutional collections and compared their anatomy to that of specimens labeled *P. gibbus* and concluded that *P. gibbus* is synonymous with *P. platessus* and variations recorded among specimens were due to intraspecific differences. However, this hypothesis has never been tested employing a robust quantitative approach. Traditional and geometric morphometrics (*Zelditch et al., 2004*) have been successfully used to interpret the patterns of morphospace occupation, quantifying the morphological diversification, solving taxonomic debates, as well as to test if morphometric variations are due to intra- or interspecific variability (*Wretman, Blom & Kear, 2016*; *Marramà & Carnevale, 2017*; *Marramà et al., 2017c*).

In this perspective, this paper aims to analyze if the morphometric variation among *Pycnodus* species of Bolca, can be related to interspecific or intraspecific variability as hypothesized by *Blot (1987)*. For this, we examined abundant *Pycnodus* specimens

from various museum collections which were labeled as either *P. apodus*, *P. platessus*, *P. gibbus* or *Pycnodus* sp. to establish whether these species separate substantially from each other in the morphospace and if morphometric and meristic data can be useful to detect significant differences between the labeled taxa. Since the studied sample had a range of specimens of different sizes, we investigated whether different shapes can be related to possible ontogenetic differences of *Pycnodus* representing different growth stages from juvenile to adult.

## The taxonomic history of *Pycnodus*

*Pycnodus* has long been used as wastebasket taxon in the study of pycnodontiforms, being used as a default name particularly for many Mesozoic taxa. Later revisions revealed said taxa to have significant morphological differences with *Pycnodus* leading to the creation of new genera. Species of pycnodontiforms previously referred to as *Pycnodus* include *Anomoeodus subclavatus* from the Maastrichtian of the Netherlands (*Agassiz, 1833*; *Davis, 1890*; *Forir, 1887*); other species of *Anomoeodus* referred to as *Pycnodus* include *A. angustus, A. muensteri, A. phaseolus, A. sculptus* (*Agassiz 1833–1844*) and *A. distans* (*Coquand, 1860*; *Sauvage, 1880*). *P. liassicus Egerton, 1855* from the Early Jurassic, of Barrow-on-Soar of Leicestershire, UK was assigned to the genus *Eomesodon* by *Woodward (1918)* and *Stemmatodus rhombus* (*Agassiz, 1833–1844*) from the Early Cretaceous of Capo d'Orlando, close to Naples, Italy was originally named *P. rhombus* (see *Heckel, 1854*). *P. flabellatum Cope, 1886* from the Cenomanian–Coniacian of Brazil was assigned to *Nursallia flabellatum* by *Blot (1987)*. The pycnodonts *P. achillis Costa, 1853, P. grandis Costa, 1853*, and *P. rotundatus Costa, 1864* are all synonymous with *Ocloedus costae* (*d'Erasmo, 1914*, *Poyato-Ariza & Wenz, 2002*). *Poyato-Ariza (2013)* revised *"Pycnodus" laveirensis Veiga Ferreira, 1961* from the Cenomanian of Lavieras, Portugal and found that due to morphological differences in characters such as absence of dermocranial fenestra, number of premaxillary teeth, contact type of arcocentra and median fin morphology, it represents a member of a different genus and consequently erected the new genus *Sylvienodus* as a replacement. An articulated specimen of *"Pycnodus"* was found in the Campanian–Maastrichtian of Nardò, Italy, which certainly represents a different pycnodont (*Taverne, 1997*). An extremely fragmentary specimen referred to as *"Pycnodus" nardoensis* from Apulia (Nardò), Italy is comprised of the anterior part of the body along with some posterior elements of the skull (*Taverne, 1997*). However, in a later study *Taverne (2003)* studied new material of this taxon, which revealed that this species does not belong to *Pycnodus* due to the possession of a narrower cleithrum and peculiar morphology of the contour scales. This new data led to the creation of the new genus *Pseudopycnodus* to allocate the Nardò material.

All other Mesozoic species of *Pycnodus* are based on isolated dentitions or teeth. The earliest records of *Pycnodus* are dentitions found in the limestones from the Upper Jurassic (Kimmeridgian) of Orbagnoux, France (*Sauvage, 1893*). Isolated teeth and an isolated vomerine dentition were referred to cf. *Pycnodus* sp. (*Goodwin et al., 1999*) from the Mugher Mudstone formation of the Tithonian. However, its identity is doubted due to the

stratigraphic position and could be attributed to *Macromesodon* (*Kriwet, 2001b*). *Pictet, Campiche & de Tribolet (1858–60)* described remains of the Early Cretaceous fish assemblages from Switzerland where three species of *Macromesodon* (*M. couloni* from the Hauterivian and Barremian, *M. cylindricus* from the Valanginian, Barremian, and Aptian and *M. obliqus* from the Albian) were all originally referred to as *Pycnodus*. Isolated dentitions belonging to "*Pycnodus*" *heterotypus* and "*Pycnodus*" *quadratifer* were reported from the Hauterivian of the Paris basin (*Cornuel, 1883*, *1886*). Several isolated teeth derived from the Cenomanian strata of the Chalk Group of southern England were attributed to *P. scrobiculatus Reuss, 1845* whose systematic affinity is still uncertain. Other teeth belonging to *P. scrobiculatus* were reported from the Turonian of northern Germany. *Roemer (1841)* described isolated remains belonging to *P. harlebeni* from the Late Cretaceous of Hilsconglomerat of Ostenvald, Germany. Another possible Portuguese representative of *Pycnodus* is reported from the Turonian of Bacarena, "*Pycnodus*" sp. aff. "*P.*" *gigas Jonet, 1964*. However, the identification of the Portuguese specimens as *Pycnodus* are uncertain and the material most likely pertains to a different pycnodont taxon (*Kriwet, 2001b*). Isolated dentitions of what were claimed to be *P. scrobiculatus*, *P. rostratus*, and *P. semilunaris* from the Turonian of Czechoslovakia (*Reuss, 1845*) should be regarded as indeterminable pycnodontids due to the lack of characters useful to determine their affinities (*Kriwet, 2001b*). Isolated teeth attributed to "*Pycnodus*" *lametae* were reported from the Maastrichtian Lameta Formation of Dongargaon, India (*Woodward, 1908*). Infratrappean and intertrappean beds of Late Cretaceous and early Palaeocene age respectively, contains "*P.*" *lametae* alongside *Pycnodus* sp. in Asifibad, India (*Prasad & Sahni, 1987*).

*Pycnodus* is the most dominant taxon of the Palaeogene pycnodont assemblages being widely distributed in shallow water contexts worldwide. The earliest record of *Pycnodus* in the Palaeogene is represented by *P. praecursor* from the Danian of Angola (*Dartevelle & Casier, 1949*) and *P.* sp. cf. *P. praecursor* from the Thanetian of Niger (*Cappetta, 1972*). *P. toliapicus* was reported from the Thanetian of Togo, Thanetian of Nigeria and the upper Palaeocene of Niger (*White, 1934*; *Kogbe & Wozny, 1979*; *Longbottom, 1984*). Several remains of isolated dentitions and teeth from the Eocene have been attributed to *Pycnodus*. These include *P. bicresta* from the northwestern Himalayan region, India (*Kumar & Loyal, 1987*; *Prasad & Singh, 1991*); *P. bowerbanki* from the Ypresian, England, middle Eocene of Mali and Ypresian of Algeria (*Longbottom, 1984*; *Savornin, 1915*); *Pycnodus* sp. cf. *P. toliapicus* from the Eocene of Katar at the Persian Gulf (*Casier, 1971*); *P. toliapicus* from the Ypresian and Lutetian of England and Lutetian of the Paris basin and Belgium (*Savornin, 1915*; *Casier, 1950*; *Taverne & Nolf, 1978*); *P. mokattamensis* from the Lutetian of Egypt (*Priem, 1897*); *P. mokattamensis* occurs alongside *P. legrandi*, *P. lemellefensis*, *P. thamallulensis*, *P. vasseuri*, and *P. pellei* from the Ypresian of Algeria (*Savornin, 1915*); *P. pachyrhinus Grey-Egerton, 1877* from the Ypresian of Kent, England; *P. funkianus Geinitz, 1883* from the Ypresian of Brunswick, Germany; *P. munieri Priem, 1902* and *P. savini Priem, 1902* from the Ypresian, France and a rather diverse assemblage from the middle Eocene of Mali which includes *P. jonesae*, *P. maliensis*, *P. munieri*, *P. variablis*, and *P. zeaformis* (*Longbottom, 1984*).
A nearly complete specimen of *P. lametae* with crushed skull and missing caudal fin was reported from the freshwater Maastrichtian of Bhatali, India close to the Dongargaon area (*Mohabey & Udhoji, 1996*). However, the assignment of the name *Pycnodus* to this fish is dubious, since it lacks the post-parietal process typical of the Pycnodontidae (J.J. Cawley, 2018, personal observation). A more complete specimen of *Pycnodus* was found in the Palaeocene rocks of Palenque, Mexico (*Alvarado-Ortega et al., 2015*), which differs from the Eocene specimens from Bolca by having a greater number of ventral and post-cloacal ridge scales, less dorsal- and anal-fin pterygiophores and a large or regular-sized posterior-most neural spine. However, due to the inadequacy of the available sample, it is not possible to determine the actual differences between the Palaeocene material from Mexico and that from the Eocene of Bolca, and for this reason this taxon is referred to as *Pycnodus* sp.

## MATERIAL AND METHODS

### Specimen sampling

We studied a selection of *Pycnodus* specimens from various museum collections, which were labeled either *P. apodus*, *P. platessus*, *P. gibbus* or *Pycnodus* sp. A total of 52 *Pycnodus* specimens from nine museum collections were used to obtain biometric information with 39 specimens from that sample being used for the geometric morphometric analysis as their higher quality preservation provided sufficient morphological information for the aim of this study (BM; Museo dei Fossili di Bolca; CM, Carnegie Museum, Pittsburgh, Pennsylvania; FMNH, Field Museum of Natural History, Chicago; MCSNV, Museo Civico di Storia Naturale di Verona; MGP-PD; Museo di Geologia e Paleontologia dell'Università di Padova; MNHN, Muséum National d'Histoire Naturelle, Paris; NHMUK, Natural History Museum of London; NHMW; Naturhistorisches Museum Wien; SNSB-BSPG, Staatliche Naturwissenshaftliche Sammlungen Bayerns-Bayerische Staatssammlung für Paläontologie und Geologie, München, Germany). For this analysis, the sample includes 17 specimens identified originally as *Pycnodus* sp., 14 specimens as *P. platessus*, six specimens as *P. gibbus*, and two specimens as *P. apodus*.

### Geometric morphometric protocol

A total of 18 landmarks, four anchor points, and 10 semi-landmarks were digitized on photos taken from the studied specimens in the corresponding collections using the software TPSdig (*Rohlf, 2005*). Landmarks indicating homologous points were selected on the basis of their possible ecological or functional role following the scheme applied in some studies (*Claverie & Wainwright, 2014*; *Tuset et al., 2014*; *Clarke, Lloyd & Friedman, 2016*; *Marramà, Garbelli & Carnevale, 2016a*, *2016b*; *Marramà et al., 2016a*; *Marramà & Carnevale, 2017*) about shape variation in modern or extinct fishes (Fig. 1). The traits used match 12 out of 17 of the landmarks that was used for 57 species of Pycnodontiformes by *Marramà et al. (2016a)*. Additional traits used here are the anterior and posterior margins of the cloaca to see if they shift significantly between morphotypes; using four landmarks around the orbit instead of one in the center to capture more

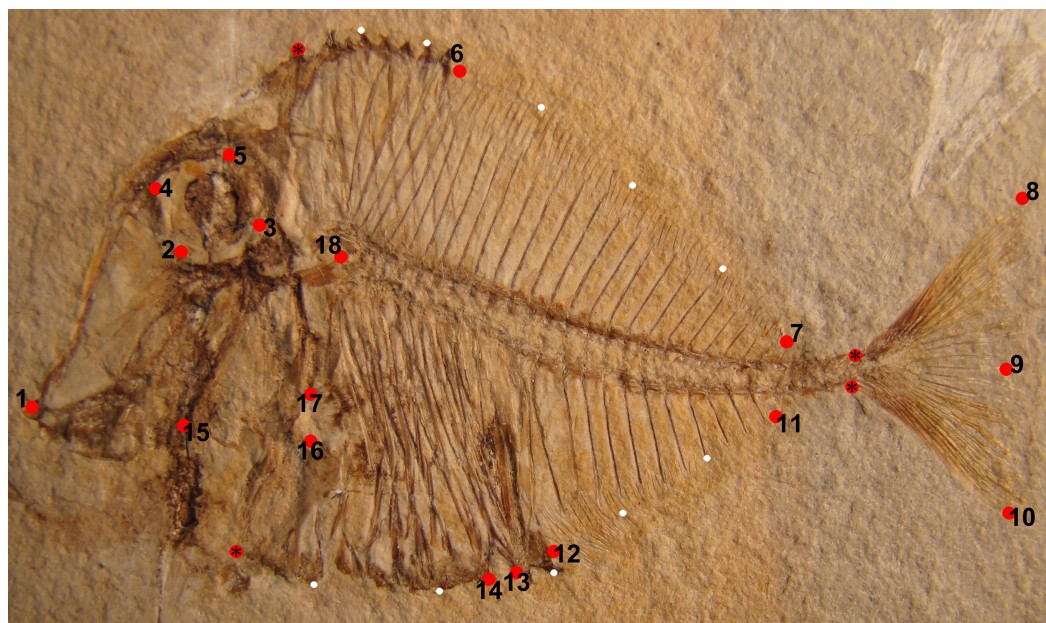

**Figure 1 Landmarks represented by red circles, which were used on *Pycnodus* (MCSNV T.998) for the geometric morphometric analysis.** These are (1) tip of premaxilla; (2) ventral-most margin of orbit; (3) posterior-most margin of orbit; (4) anterior-most margin of orbit; (5) dorsal-most margin of orbit; (6) first dorsal pterygiophore; (7) last dorsal pterygiophore; (8) tip of dorsal lobe of caudal fin; (9) medial convex margin of caudal fin; (10) tip of ventral lobe of caudal fin; (11) final anal pterygiophore; (12) first anal pterygiophore; (13) posterior cloacal scale; (14) anterior cloacal scale; (15) joint between quadrate and prearticular; (16) ventral-most concave margin of cleithrum accommodating pectoral fin; (17) dorsal-most concave margin of cleithrum accommodating pectoral fin; (18) Point of contact between neurocranium and vertebral column. Red circles marked with an asterisk are anchor points for the semi-landmarks. The semi-landmarks are represented by small white circles and are split into two sets; the first set consists of seven semi-landmarks between the tip of the dermosupraoccipital and the base of the first principal caudal fin ray; the second set has an additional seven semi-landmarks between the base of the ventral-most principal caudal fin ray and the antero-ventral corner of the cleithrum. Photo credit: Jürgen Kriwet.

precisely the variability surrounding the orbit; not using the insertion of the pelvic fin as this character was rarely preserved in our specimens; the use of two landmarks for the cleithrum to capture variability in position and size of the pectoral fin instead of using just the one landmark for the insertion of the first pectoral fin ray due to the poor preservation of the pectoral fins in many specimens in contrast to the concave notch in the cleithrum.

The landmark coordinates were translated, rotated and scaled at unit centroid size by applying a Generalized Procrustes Analysis (GPA) to minimize the variation caused by size, orientation, location and rotation (*Rohlf & Slice, 1990*; *Zelditch et al., 2004*). The GPA was performed using the TPSrelw software package (*Rohlf, 2003*) and a principal component analysis was performed on Procrustes coordinates to obtain the Relative Warp (RW). Shape changes were shown along the axes using deformation grid plots. Missing values are replaced using the algorithm "Mean value imputation" (*Hammer, Harper & Ryan, 2001*).
Two non-parametric tests were performed to analyze the quantitative morphospace occupation of our *Pycnodus* specimens. In order to assess the degree of overlap between morphospaces, an analysis of similarities (ANOSIM) (*Clarke, 1993*) was performed on the entire dataset of standardized morphometric and meristic parameters. PERMANOVA (*Anderson, 2001*) was used to test similarities of in-group centroid position between the different groups representing a species of *Pycnodus*. Euclidean distances are the distance measure chosen for both tests. All statistical analyses were performed in PAST 3.18 (*Hammer, Harper & Ryan, 2001*).

Since the studied specimens vary greatly in size (smallest being 4.0 cm and largest being 30.6 cm) we also investigated whether size could be correlated with shape change in *Pycnodus* and enable us to see whether and how body shape changes throughout ontogeny. To analyze the relationship between size and shape, we performed a partial least square analysis (PLS) using the software TPSpls (*Rohlf & Corti, 2000*). Alpha (level of significance) was set to 0.05.

## Biometric analyses

We used 11 meristic counts (number of vertebrae, ribs, scale bars, paired fin rays, median fin rays, median fin pterygiophores, caudal fin rays, and arcocentra interdigitations) and 19 measurements (see Supplementary Material) in order to capture morphological variability, to test the homogeneity of the sample, and confirming its assignment to a single morphotype. Histograms were used to illustrate the variation of morphometric and meristic data in order to ascertain if more than one morphotype of *Pycnodus* could be identified. Histograms can be problematic in accurately capturing the distribution of data (*Salgado-Ugarte et al., 2000*) so we also used Kernel density estimators to determine the presence of a normal (Gaussian) distribution of the meristic data. Least squares regression was used to obtain the relationship between standard length (SL) and all other morphometric variables. Specimens of possible additional taxa were indicated by the presence of statistical outliers from the regression line (*Simon et al., 2010*) and will require additional scrutiny in order to truly differentiate the outlier from all other specimens. The linear regression results were shown using scatterplots. Log-transformed data were used to perform the least squares regression in order to determine the degree of correlation between the SL and all other morphometric variables.

## RESULTS

### Geometric morphometrics

The RW analysis produced 38 RWs with the first three axes together explaining about 73% of the total variation. Figures 2 and 3 show that there is significant overlap between the morphospaces of the *Pycnodus* taxonomic groups and the thin plate splines show the changes in shape along the axes. Negative values on RW1 (56.3% explained) are related to *Pycnodus* specimens with large orbits and deep bodies while positive scores identify *Pycnodus* with reduced orbits and elongated bodies. Negative values of RW2 (10.3% explained) show specimens having the pectoral fin with a wide base moved higher up the
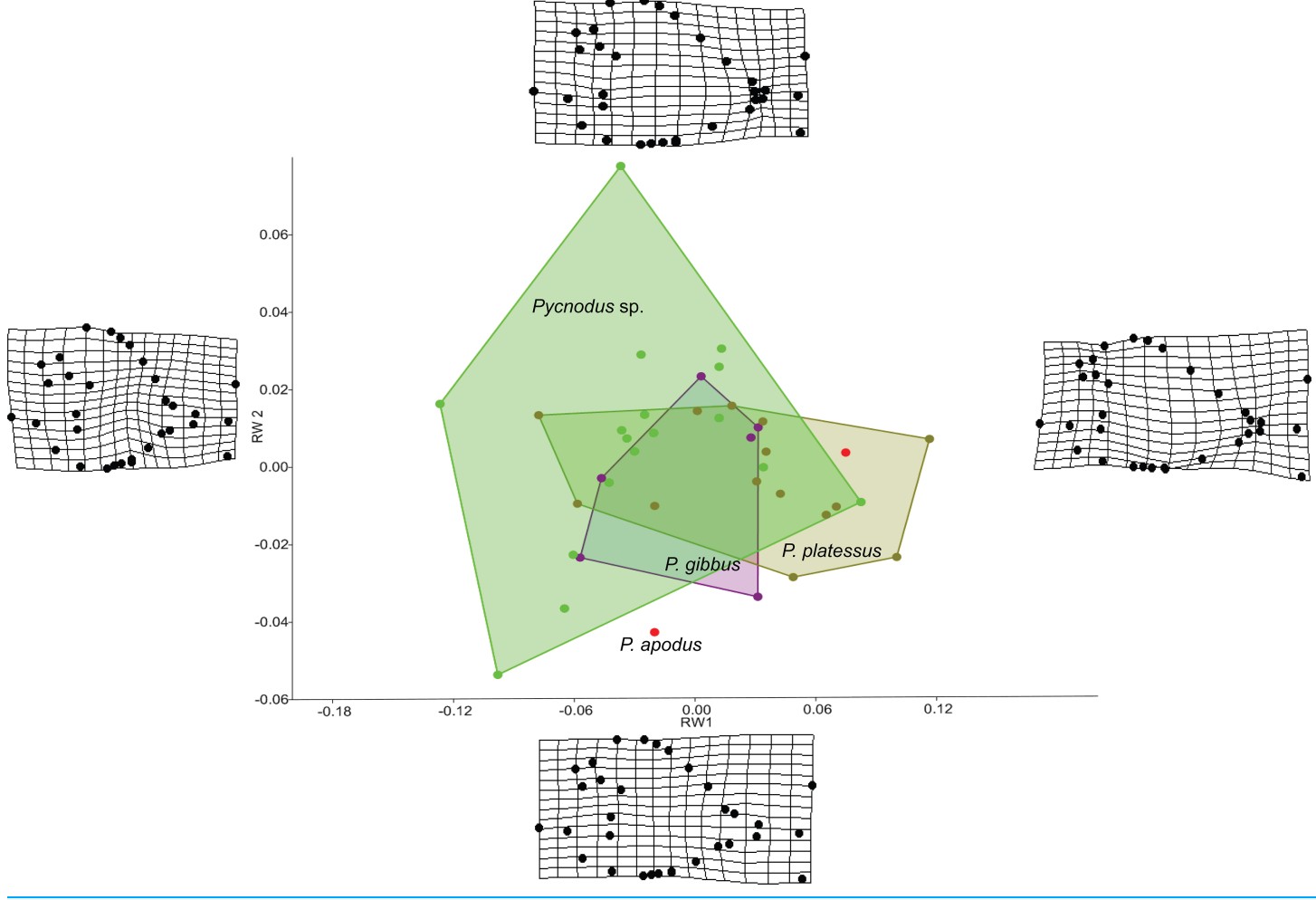

**Figure 2 Morphospace of *Pycnodus* on the first two RW axes together accounting for about 66% of the overall shape variation.** Deformation grids illustrate the shapes lying at extreme values along each axis.

body alongside a long caudal peduncle (Fig. 2). Conversely, on positive scores of RW2 lie specimens with pectoral fin with a narrower base located more ventrally on the body alongside a small caudal peduncle. The negative values of RW3 (5.9% explained) show the skull becoming deeper and more elongated with the dermosupraoccipital in particular reaching far back (Fig. 3). The body becomes shallower near the caudal peduncle with the cloaca shifting posteriorly, as well as the dorsal apex. Positive scores of RW3 are related to a shorter and shallower skull with the body becoming deeper close to the caudal peduncle and the anterior shift in the cloaca with the body becoming deeper just anterior to the cloaca. The dorsal apex shifts forward in position.

Analysis of similarities performed on the first three axes suggests that there is strong overlap between groups, showing they are barely distinguishable from each other ($r$-value is 0.10 and $p > 0.05$; see Table 1), except for a single pairwise comparison between *Pycnodus* sp. and *P. platessus* ($p < 0.05$). The PERMANOVA suggests the same trend (Table 2), showing that group centroids are not significantly different on each pairwise

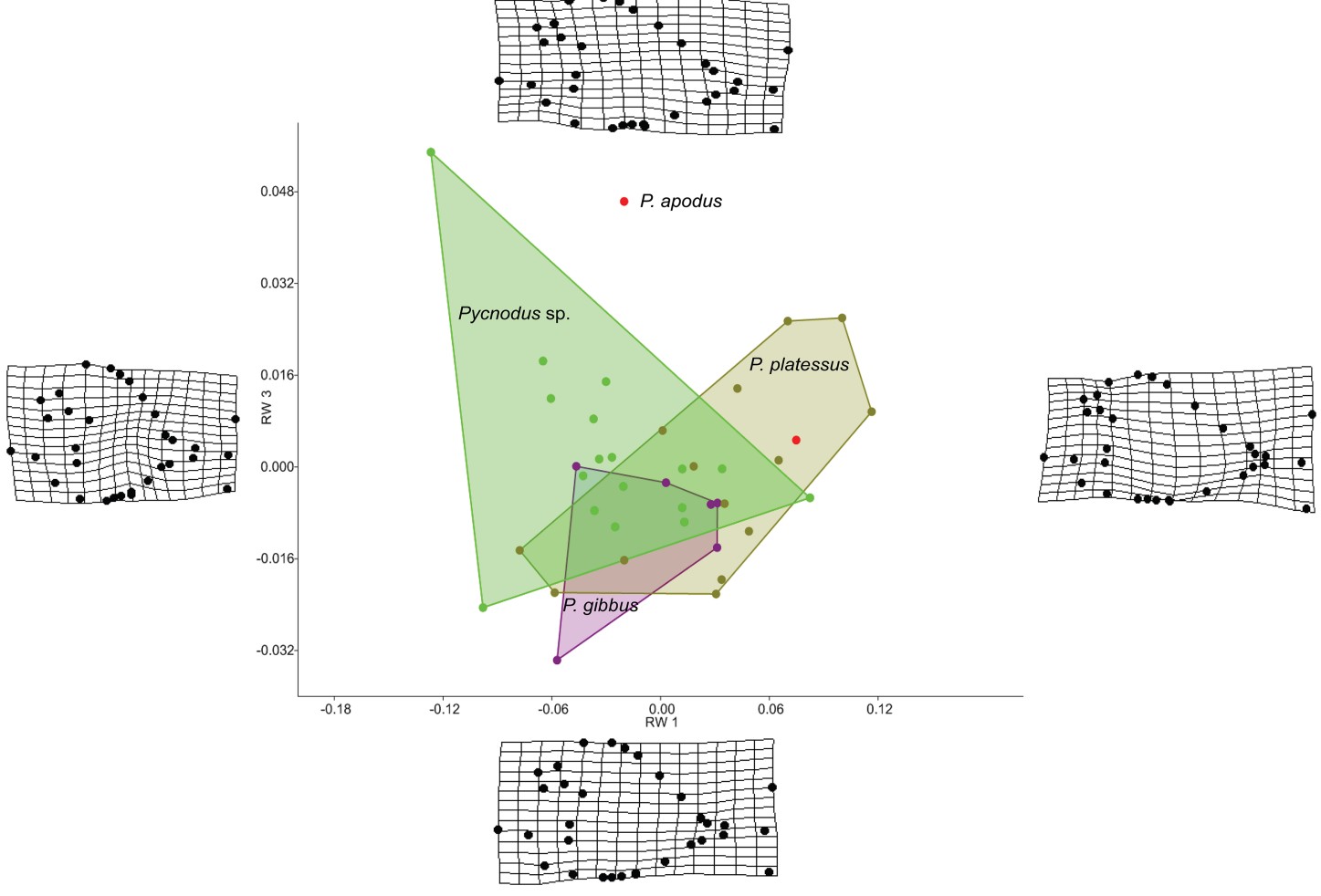

**Figure 3 Morphospace of *Pycnodus* showing RW 1 on the *x*-axis and RW 3 on *y*-axis, the latter accounting for 6% of the overall shape variation.** Deformation grids illustrate the shapes lying at extreme values along each axis.

**Table 1 ANOSIM results.**

| ANOSIM | *P. apodus* | *P. gibbus* | *P. platessus* | *Pycnodus* sp. |
|---|---|---|---|---|
| *P. apodus* | 0 | 0.3583 | 0.7879 | 0.1717 |
| *P. gibbus* | 0.3583 | 0 | 0.3411 | 0.4755 |
| *P. platessus* | 0.7879 | 0.3411 | 0 | 0.0389 |
| *Pycnodus* sp. | 0.1717 | 0.4755 | 0.0389 | 0 |

**Note:**
    *r*-value is 0.10 and *p*-value is 0.06.

comparison (*f*-value is 2.83), except between *Pycnodus* sp. and *P. platessus* ($p < 0.05$) which lends significance to the overall *p*-value (<0.05). Significant differences detected between *Pycnodus* sp. and *P. platessus* can be explained with the fact that the indeterminate *Pycnodus* specimens show a wide range of morphologies, with the extreme shapes ranging from negative to positive values of all the first three axes.

**Table 2  PERMANOVA results.**

| PERMANOVA | *P. apodus* | *P. gibbus* | *P. platessus* | *Pycnodus* sp. |
|---|---|---|---|---|
| *P. apodus* | 0 | 0.3228 | 0.5671 | 0.1586 |
| *P. gibbus* | 0.3228 | 0 | 0.2358 | 0.2876 |
| *P. platessus* | 0.5671 | 0.2358 | 0 | 0.0048 |
| *Pycnodus* sp. | 0.1586 | 0.2876 | 0.0048 | 0 |

**Note:**
*f*-value is 2.83 and *p*-value is 0.03.

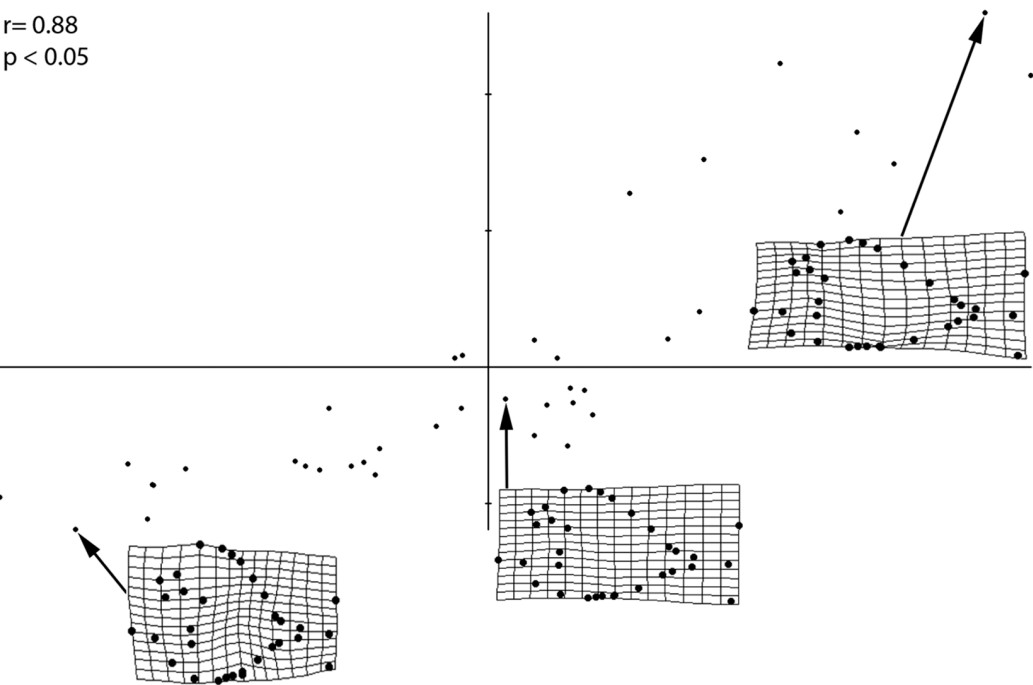

**Figure 4  PLS analysis showing a correlation of morphometric variation with size.** Smallest, medium sized, and largest specimens are used to represent the juvenile, small adult, and large adult stages, respectively. Significance of this correlation is shown by the *r* and *p*-values. Smallest specimen is 4.02 cm, medium sized specimen is 10.6 cm, largest specimen is 30.6 cm.

   The PLS performed on the entire sample (Fig. 4) revealed a strong and significant correlations between size and shape ($r = 0.88$; $p < 0.05$), therefore suggesting that different shapes of the individuals are related to changes in shape of different ontogenetic stages. Small-sized individuals are associated with larger orbits, deeper skull and body shape, long skull, higher position of pectoral fin and a wide, indistinct caudal peduncle that is in distant proximity to both medial fins. Larger individuals, on the other hand, have a reduced orbit, shallower skull and body depth, shorter skull, lower position of pectoral fin and narrow caudal peduncle in close proximity to both medial fins. The PLS analysis therefore suggests that the morphological variations of the orbit, body depth and caudal peduncle are strongly related to ontogeny.

**Table 3 Measurements as percentage of SL (mean values in parentheses) used for identifying *Pycnodus apodus*.**

| Morphometric character | Measurements in % of SL |
|---|---|
| Head length | 27.9–32.9 (30.4) |
| Head depth | 48.5–57.7 (53.1) |
| Maximum body depth | 55.6–65.1 (60.8) |
| Pectoral fin base | 6.5–9.2 (8.1) |
| Dorsal fin base | 37.4–44.3 (40.9) |
| Anal fin base | 25.3–29.4 (27.8) |
| Caudal peduncle depth | 3.8–5.1 (4.6) |
| Caudal peduncle length | 13.6–15.7 (14.7) |
| Caudal fin span | 32.9–38.6 (35.9) |
| Prepectoral distance | 28.1–30.7 (29.6) |
| Predorsal distance | 41.9–48.3 (45.2) |
| Prepelvic distance | 48.6–52.7 (50.4) |
| Preanal distance | 56.9–60.3 (58.6) |
| Preorbital distance | 9.9–14.4 (12.3) |
| Postorbital length | 5.4–8.3 (7.1) |
| Orbit diameter | 9.3–12.5 (11.0) |
| Lower jaw | 12.5–16.5 (14.7) |

**Note:**
Range of measurements are represented by the 25th and 75th percentile.

**Table 4 Mean meristic values used for identifying *Pycnodus apodus*.**

| Meristic character | Mean meristic value |
|---|---|
| Vertebrae | 24–26 (25) |
| Rib pairs | 10–12 (11) |
| Scale bars | 8–10 (9) |
| Dorsal fin rays | 54–60 (56) |
| Anal fin rays | 42–48 (45) |
| Pectoral fin rays | 30–40 (35) |
| Dorsal fin pterygiophores | 53–60 (56) |
| Anal fin pterygiophores | 41–41 (45) |
| Caudal fin rays | 25–34 (30) |

**Note:**
Range of meristic counts are represented by the 25th and 75th percentile. Mean meristic value in parentheses.

## Biometric analyses

Morphometrics and meristic counts for all the studied specimens are given in Tables 3 and 4, respectively and mean biometric parameters are given in Table 5. Most of the histograms based on meristic counts (Fig. 5) do not show a normal (Gaussian) distribution due to the small sample size being unable to detect significant high frequency of mean values that might have suggested a Gaussian curve (*De Baets, Klug & Monnet, 2013*), with intermediate states dominating and extreme states being rare. The linear

**Table 5 Mean morphometric and meristic data for the examined specimens of *Pycnodus*.**

| Morphometric/meristic data | Min | Max | Mean | Median | Variance | Standard deviation | 25th percentile | 75th percentile |
|---|---|---|---|---|---|---|---|---|
| Standard length | 2.9 | 27.7 | 11.1 | 8.8 | 46.7 | 6.8 | 5.9 | 16.4 |
| Head length | 1.1 | 7.1 | 3.3 | 2.8 | 2.9 | 1.7 | 2.0 | 4.6 |
| Head depth | 2.0 | 11.6 | 5.6 | 4.4 | 7.7 | 2.8 | 3.5 | 7.8 |
| Maximum body depth | 2.1 | 13.4 | 5.8 | 4.9 | 8.4 | 2.9 | 3.8 | 7.4 |
| Pectoral fin base | 0.2 | 1.8 | 0.8 | 0.7 | 0.2 | 0.4 | 0.5 | 1.1 |
| Dorsal fin base | 1.1 | 12.5 | 4.9 | 3.7 | 10.5 | 3.2 | 2.4 | 6.3 |
| Anal fin base | 0.7 | 9.6 | 3.4 | 2.5 | 5.6 | 2.4 | 1.6 | 5.0 |
| Caudal peduncle depth | 0.2 | 1.2 | 0.5 | 0.4 | 0.1 | 0.3 | 0.3 | 0.6 |
| Caudal peduncle length | 0.6 | 3.7 | 1.6 | 1.3 | 0.8 | 0.9 | 1.0 | 2.4 |
| Caudal fin span | 0.9 | 10.7 | 4.1 | 3.0 | 6.7 | 2.6 | 2.2 | 6.9 |
| Prepectoral distance | 1.1 | 7.2 | 3.1 | 2.8 | 2.5 | 1.6 | 1.9 | 4.0 |
| Predorsal distance | 1.6 | 11.0 | 5.0 | 4.2 | 7.4 | 2.7 | 2.9 | 7.6 |
| Prepelvic distance | 1.7 | 12.4 | 5.3 | 4.3 | 8.9 | 3.0 | 3.2 | 6.4 |
| Preanal distance | 2.2 | 14.2 | 6.6 | 5.4 | 12.8 | 3.6 | 3.7 | 9.3 |
| Preorbital distance | 0.3 | 4.1 | 1.4 | 1.1 | 1.0 | 1.0 | 0.8 | 1.9 |
| Postorbital length | 0.3 | 1.7 | 0.7 | 0.6 | 0.1 | 0.3 | 0.5 | 0.8 |
| Orbit diameter | 0.4 | 2.2 | 1.1 | 1.0 | 0.2 | 0.4 | 0.8 | 1.3 |
| Lower jaw | 0.5 | 4.6 | 1.7 | 1.3 | 1.1 | 1.0 | 0.9 | 2.4 |
| Vertebrae | 23 | 27 | 25.1 | 25 | 1.4 | 1.2 | 24 | 26 |
| Rib pairs | 9 | 13 | 11.1 | 11 | 1.1 | 1.1 | 10 | 12 |
| Scale bars | 7 | 11 | 8.7 | 8 | 0–9 | 1.0 | 8 | 10 |
| Dorsal fin rays | 46 | 66 | 56.4 | 56 | 18.2 | 4.3 | 54 | 60 |
| Anal fin rays | 37 | 52 | 45.0 | 45 | 14.5 | 3.8 | 42 | 47.8 |
| Pectoral fin rays | 24 | 47 | 35.2 | 35.5 | 43.9 | 6.6 | 30.3 | 39.8 |
| Pelvic fin rays | 3 | 5 | 4.3 | 4 | 0.6 | 0.8 | 4 | 5 |
| Dorsal fin pterygiophores | 38 | 65 | 55.8 | 57 | 30.5 | 5.5 | 52.8 | 60 |
| Anal fin pterygiophores | 39 | 58 | 44.8 | 45 | 16.3 | 4.0 | 41 | 47 |
| Caudal fin rays | 22 | 43 | 29.5 | 29 | 35.8 | 6.0 | 24.5 | 33.5 |
| Arcocentra interdigitations | 2 | 3 | 2 | 2 | 0 | 0.2 | 2 | 2 |

regression performed on morphometric characters (Fig. 6) shows that all specimens fit within the cloud of points near the regression line and that no particular specimens of *Pycnodus* deviates from this line. Variation in meristic values and the few outliers in partial least square regression analyses have been interpreted here as measurement errors due to incomplete preservation of some structures due to taphonomy or incomplete mineralization in juvenile individuals. The high values of the coefficient of determination ($r^2$) ranging from 0.76 to 0.99 (Table 6) indicate a high degree of positive correlation between SL and each morphometric character. Linear regression analysis also revealed the highly significant relationship between the SL and all morphometric characters ($p < 0.001$). Neither morphometric nor meristic characters are therefore useful in determining two or more different morphologically identifiable species within *Pycnodus*,

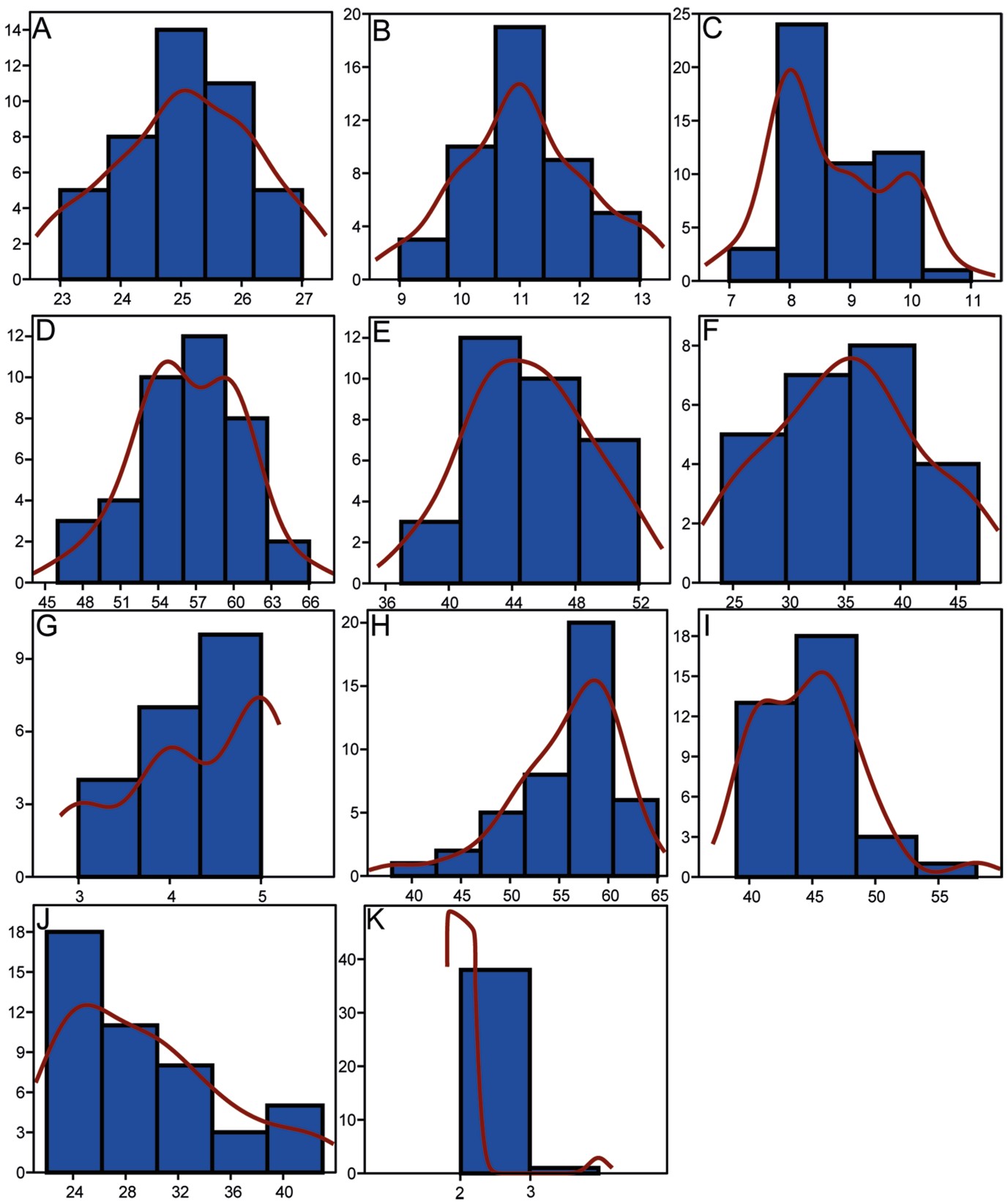

**Figure 5 Histograms showing the distributions of meristic characters of *Pycnodus*.** The *x*-axis represents the number of elements and the *y*-axis the relative frequency. Red curved line is the Kernel density estimator which measures the normality of each sample which reveals that there is a non-Gaussian distribution among all the samples. (A) Vertebrae. (B) Rib pairs. (C) Scale bars. (D) Dorsal fin rays. (E) Anal fin rays. (F) Pectoral fin rays. (G) Pelvic fin rays. (H) Dorsal fin pterygiophores. (I) Anal fin pterygiophores. (J) Caudal fin rays. (K) Arcocentra interdigitations.

strongly supporting *Blot's (1987)* hypothesis that only one species (*P. apodus*; see also below) is present in the Bolca Lagerstätte.

## DISCUSSION

### Intraspecific variation of *Pycnodus apodus*

The results demonstrate that all *Pycnodus* species cannot be separated morphologically using the morphometric traits used herein in a quantitative approach, supporting the intraspecific variation hypothesis of *Blot (1987)*. *P. gibbus* is a problematic taxon to identify due to *Heckel (1856)* not mentioning exactly which specimen he used to designate the specific name for *P. gibbus*. *Blot (1987)* mentions that Heckel worked on specimens from the NHMW in order to erect *P. gibbus*. However, such specimens could not be found and so the holotype still remains unknown. However, *Heckel (1856*, plate 8*)* does illustrate a specimen of *P. gibbus* and it conforms with what we have found to be the juvenile morphotype in our sample lending credence to the hypothesis by *Agassiz (1833–1844)* that the specimens he studied were specifically the juvenile of *P. platessus*. One of the characters separating *P. gibbus* from *P. platessus* (*Heckel, 1856*, plate 8, Fig. 4) is the number of interdigitations between vertebrae (*P. gibbus*: two; *P. platessus*: three-four). However, a survey of the vertebral column of all our specimens reveals two to be the predominant number of interdigitations, including specimens labeled *P. platessus* and *P. apodus*. Apart from specimens where the degree of preservation was insufficient to do a count, only one specimen (MGP-PD 8868C) has three interdigitations which we ascertain to be due to intraspecific variation. *Blot (1987*, Table 6*)* also did not see any difference in the number of interdigitations between *P. gibbus* and *P. platessus*.

As suggested by *Grande & Young (2004)*, ontogenetic variation of morphological characters actually represents a primary source of intraspecific variation; this is confirmed by our analysis, specifically by the morphological changes mostly occurring along RW1 in the morphospace that are related to ontogeny and the very significant results deriving from the PLS analysis. The unimodal (Gaussian) distribution cannot be seen in most of the meristic data, as revealed by the Kernel density estimator on the frequency histograms (Fig. 5), due to the fact that the sample is too small to detect high frequency of mean values (*De Baets, Klug & Monnet, 2013*). However, a few meristic characters reveal a domination of intermediate values and comparably rare extremes, which is typical of a homogenous population. Furthermore, the linear regression showed a significant dependence between SL and all morphometric variables, therefore suggesting that morphometric characters are not useful to distinguish different taxa. Meristic and morphometric data seem to show that all specimens studied belong to a single taxonomic entity (see *Dagys, Bucher & Weitschat, 1999*; *Dagys, 2001*; *Weitschat, 2008*; *Marramà & Carnevale, 2015a*; *Sferco, López-Arbarello & Báez, 2015*).

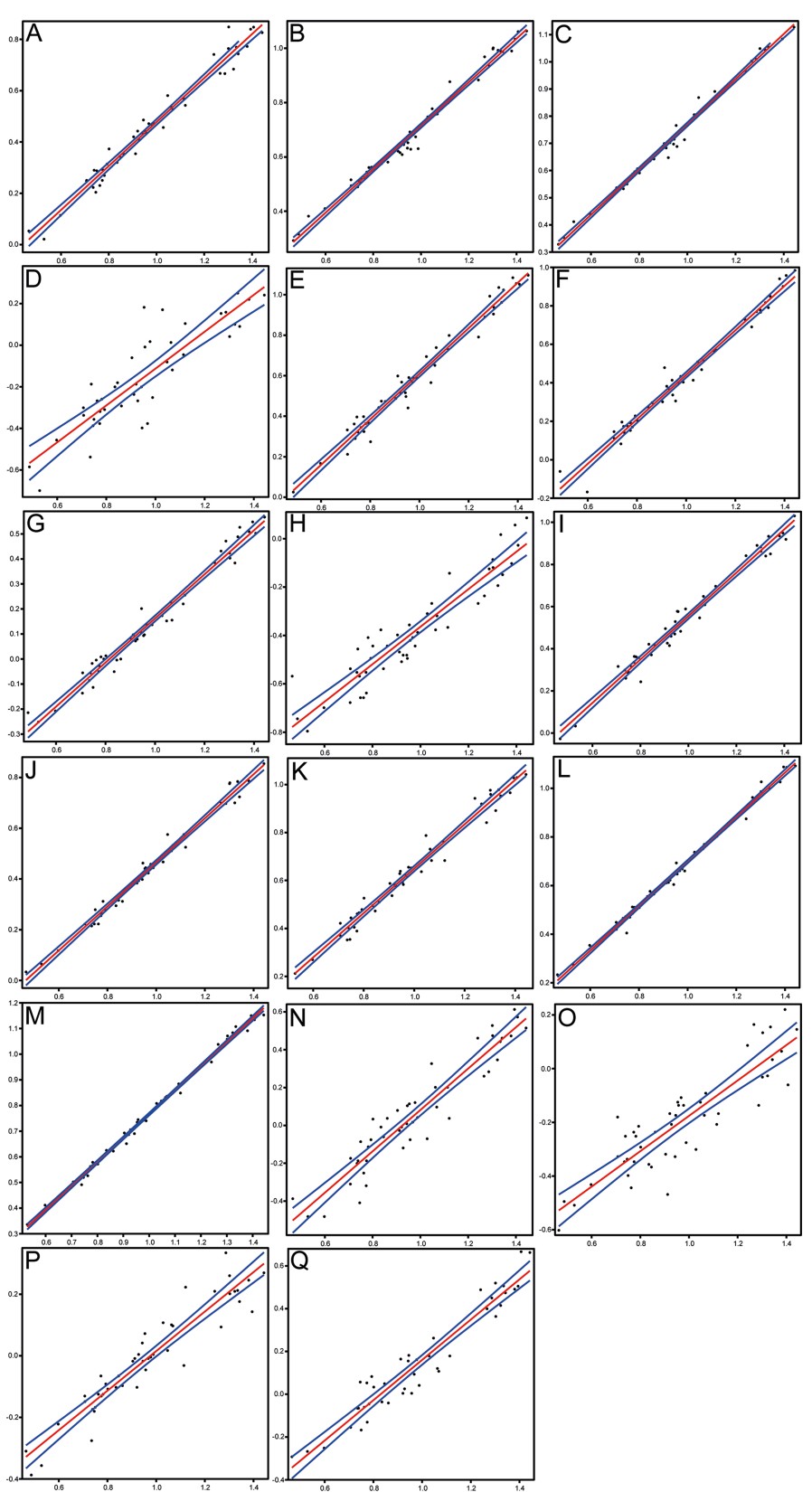

**Figure 6 Scatterplots and regression lines with 95% confidence bands of the relationships between each morphometric character and the standard length of *Pycnodus*.** (A) Head length. (B) Head depth. (C) Maximum body depth. (D) Pectoral fin base. (E) Dorsal fin base. (F) Anal fin base. (G) Caudal peduncle length. (H) Caudal peduncle depth. (I) Caudal fin span. (J) Prepectoral distance. (K) Predorsal distance. (L) Prepelvic distance. (M) Preanal distance. (N) Preorbital length. (O) Postorbital length. (P) Orbit diameter. (Q) Lower jaw length.

**Table 6 Relationships between morphometric characters and standard length using least squares regression for *Pycnodus*.**

| Variable character log $(y)$ | Slope $(a)$ | Intercept $(b)$ | Coefficient of determination $(r^2)$ | 95% CI on $a$ | | 95% CI on $b$ | |
|---|---|---|---|---|---|---|---|
| Head length | 0.86 | −0.38 | 0.97 | 0.80 | 0.90 | −0.42 | −0.33 |
| Head depth | 0.80 | −0.09 | 0.98 | 0.77 | 0.83 | −0.11 | −0.06 |
| Maximum body depth | 0.83 | −0.06 | 0.99 | 0.81 | 0.85 | −0.08 | −0.04 |
| Pectoral fin base | 0.89 | −1.00 | 0.76 | 0.77 | 0.99 | −1.11 | −0.88 |
| Dorsal fin base | 1.12 | −0.51 | 0.97 | 1.07 | 1.17 | −0.56 | −0.46 |
| Anal fin base | 1.16 | −0.71 | 0.97 | 1.09 | 1.22 | −0.78 | −0.64 |
| Caudal peduncle depth | 0.77 | −1.13 | 0.89 | 0.68 | 0.87 | −1.23 | −1.05 |
| Caudal peduncle length | 0.91 | −0.75 | 0.97 | 0.85 | 0.97 | −0.81 | −0.69 |
| Caudal fin span | 1.04 | −0.49 | 0.98 | 1.00 | 1.09 | −0.54 | −0.45 |
| Prepectoral distance | 0.87 | −0.40 | 0.98 | 0.83 | 0.90 | −0.43 | −0.36 |
| Predorsal distance | 0.91 | −0.26 | 0.98 | 0.86 | 0.95 | −0.30 | −0.21 |
| Prepelvic distance | 0.92 | −0.22 | 0.99 | 0.89 | 0.94 | −0.24 | −0.19 |
| Preanal distance | 0.93 | −0.17 | 0.99 | 0.91 | 0.95 | −0.19 | −0.14 |
| Preorbital distance | 1.09 | −1.01 | 0.89 | 0.99 | 1.20 | −1.12 | −0.90 |
| Postorbital length | 0.66 | −0.83 | 0.78 | 0.56 | 0.76 | −0.93 | −0.74 |
| Orbit diameter | 0.64 | −0.63 | 0.89 | 0.57 | 0.71 | −0.69 | −0.56 |
| Lower jaw | 0.94 | −0.78 | 0.92 | 0.87 | 1.02 | −0.86 | −0.70 |

Figure 7 shows some notable differences between the juvenile and larger specimens including the degree of ossification, particularly in the skull and caudal fin, being reduced in juvenile in comparison to adults and the notochord not being surrounded by arcocentra in juveniles whereas it is completely enclosed in adults. The so-called gibbosity that *Heckel (1856)* used to distinguish *P. gibbus* from *P. platessus* is formed by the angle of the anterior profile and the axis of the body. This angle decreases in larger specimens of *Pycnodus* from 70° to 55° (*Blot, 1987*) due to the skull roof moving posteriorly during growth revealing that this character probably does not denote a species but a growth stage within a single species. The high vertebrae length/body depth ratio said to be another indicator of *P. gibbus* is something that also decreases during growth. When Blot plotted all *Pycnodus* specimens onto a growth curve (*Blot, 1987*, fig. 32) *P. gibbus* fitted into the curve neatly on the lower end of the growth curve.

Differences in meristic counts (Table 7) are suggestive of intraspecific variation as seen in other fossil actinopterygians such as Sinamiidae from the Late Jurassic (*Su, 1973*; *Zhang & Zhang, 1980*) and Early Cretaceous (*Stensiö, 1935*); Palaeosconiformes from the Triassic (*Lehman, 1952*); Parasemionotidae from the Early Triassic (*Olsen, 1984*)

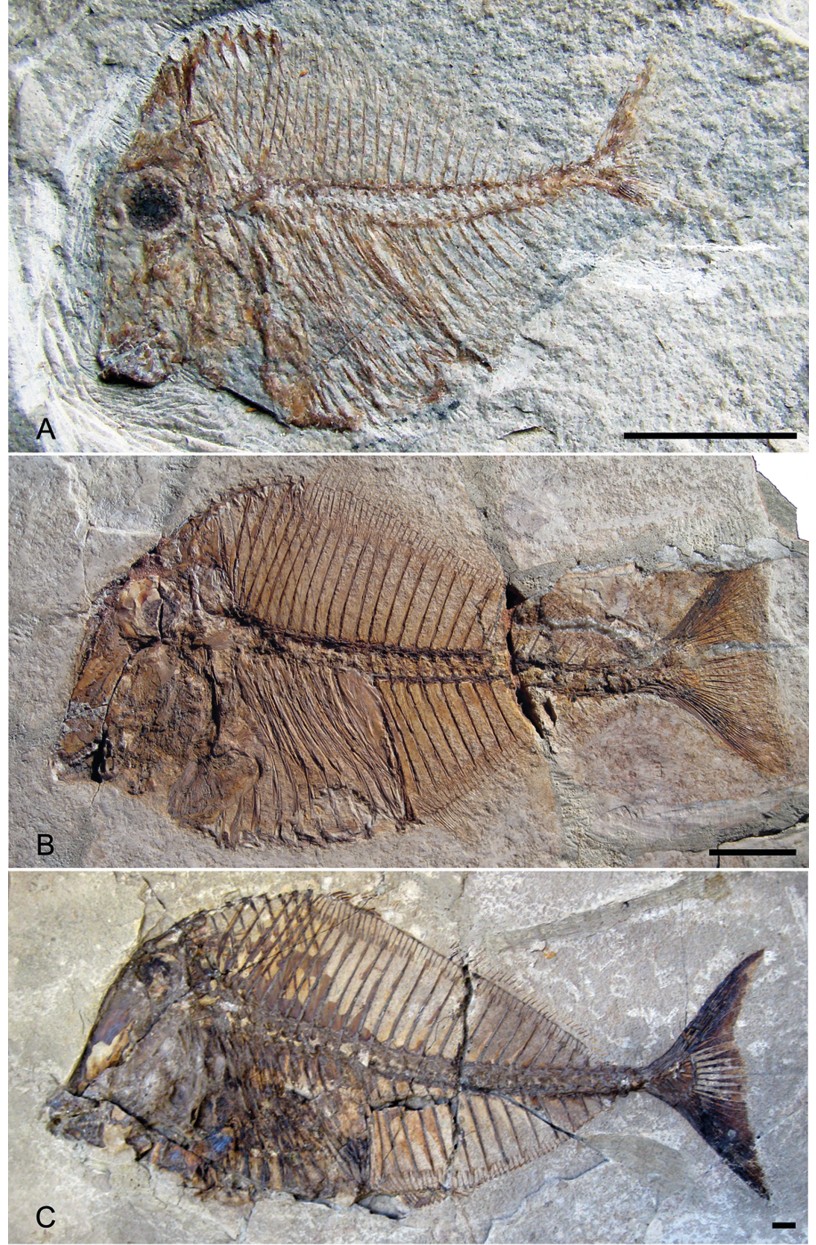

**Figure 7 Ontogenetic series of *Pycnodus*.** (A) Juvenile 4.02 cm (MCSNV T.309). (B) Small adult 13.25 cm (BSPG AS I 1208). (C) Large adult 30.61 cm (BSPG AS I 1209). Scale bar for all specimens equals 1 cm. Photo credit: Jürgen Kriwet.

Teleosteomorpha from the Middle to Late Triassic (*Tintori, 1990*); *Bobasatraniiformes* from the Middle Triassic (*Bürgin, 1992*) Paramblypteidae from the Early Permian (*Dietze, 1999*, *2000*) Dapediidae from the Early Jurassic (*Thies & Hauff, 2011*); stem Actinopteri from the Middle Triassic (*Xu, Shen & Zhao, 2014*); stem Teleostei from the Middle Triassic (*Tintori et al., 2015*); Pachycormiformes from the Early Jurassic (*Wretman, Blom & Kear, 2016*); and the *incertae sedis* genus *Teffichthys* from the Early Triassic *Marramà et al., 2017c*). The analysis of the morphological variability of *Pycnodus*,

**Table 7 Meristic counts of *Pycnodus*.**

| Species name on museum label | Specimen no. | No. of vertebrae | Rib pairs | No. of scale bars | Dorsal fin rays | Anal fin rays | Pectoral fin rays | Pelvic fin rays | Dorsal fin pterygiophores | Anal fin pterygiophores | Caudal fin rays | Arcocentra interdigitations | Museum |
|---|---|---|---|---|---|---|---|---|---|---|---|---|---|
| *Pycnodus* sp. | 12058 | 26 | 13 | 8 | 60 | ? | 39 | ? | 57 | ? | 32 | 2 | MGP-PD |
| *Pycnodus* sp. | 12059 | 25 | ? | 9 | 52 | ? | 44 | ? | 53 | ? | 29 | 2 | MGP-PD |
| *Pycnodus* sp. | 12808 | 24 | 12 | 8 | ? | ? | 44 | ? | ? | 40 | 26 | 2 | MGP-PD |
| *Pycnodus* sp. | 12809 | 25 | ? | 8 | 56 | 42 | 28 | ? | 56 | 44 | 30 | 2 | MGP-PD |
| *Pycnodus* sp. | 26968 | ? | 12 | 8 | ? | 43 | 33 | ? | ? | 40 | ? | 2 | MGP-PD |
| *Pycnodus* sp. | 26969 | 25 | 11 | 10 | 55 | 46 | 25 | ? | 58 | 44 | 30 | 2 | MGP-PD |
| *Pycnodus platessus* | 1853.XXVI.i.a/b | 25 | 10 | 9 | 61 | 46 | 47 | 5 | ? | 47 | ? | 2 | NHMW |
| *Pycnodus platessus* | 1855.VI.75 | 23 | 10 | 8 | 54 | 42 | 38 | 3 | 54 | 40 | 24 | 2 | NHMW |
| *Pycnodus platessus* | 6880Z | 25 | 13 | 10 | ? | ? | 36 | ? | 48 | ? | 22 | 2 | MGP-PD |
| *Pycnodus gibbus* | 7433C | 25 | 11 | 9 | ? | ? | ? | 4 | 52 | ? | 25 | 2 | MGP-PD |
| *Pycnodus platessus* | 8867C | 26 | 11 | 8 | 56 | ? | ? | ? | 57 | 46 | 23 | ? | MGP-PD |
| *Pycnodus platessus* | 8868C | ? | 13 | 7 | 54 | 49 | ? | ? | 60 | 46 | 25 | 3 | MGP-PD |
| *Pycnodus platessus* | A.III.a.S.48 | 24 | 11 | 8 | 56 | 45 | ? | ? | 59 | 46 | 28 | 2 | NHMW |
| *Pycnodus platessus* | BMNH 38000 | 26 | 10 | 8 | 66 | ? | ? | 5 | 65 | 48 | 24 | ? | BMNH |
| *Pycnodus gibbus* | BMNH P.11992 | 27 | 11 | 10 | 55 | ? | ? | 3 | 60 | 46 | 26 | 2 | BMNH |
| *Pycnodus gibbus* | BMNH P.1632/ P.3760 | 27 | 11 | 11 | 49 | ? | ? | 3 | 53 | ? | 31 | 2 | BMNH |
| *Pycnodus platessus* | BMNH P.1633 | 25 | 11 | 9 | 59 | 47 | 31 | 5 | 62 | 45 | 29 | 2 | BMNH |
| *Pycnodus gibbus* | BMNH P.17025 | 24 | 10 | 10 | 52 | 41 | 30 | ? | 49 | 39 | 27 | 2 | BMNH |
| *Pycnodus gibbus* | BMNH P.4386 | ? | 12 | 10 | ? | ? | 46 | 5 | ? | ? | 43 | 2 | BMNH |
| *Pycnodus gibbus* | BMNH P.44519 | 26 | 12 | 8 | 61 | 50 | 35 | 3 | 63 | 44 | 36 | 2 | BMNH |
| *Pycnodus gibbus* | BMNH P.44520 | 26 | 10 | 9 | 62 | 39 | ? | ? | 60 | ? | 37 | 2 | BMNH |
| *Pycnodus platessus* | BMNH P.7459 | ? | 10 | 8 | 63 | 45 | 36 | 5 | 59 | 51 | 34 | 2 | BMNH |
| *Pycnodus apodus* | Bol 126/127 | 26 | 11 | 10 | 52 | ? | 40 | 5 | ? | ? | 33 | 2 | MNHN |
| *Pycnodus apodus* | Bol 130/131 | ? | 10 | 9 | ? | ? | ? | ? | ? | ? | ? | 2 | MNHN |
| *Pycnodus apodus* | Bol 134/135 | 25 | 11 | 10 | 59 | 52 | ? | 5 | 61 | 48 | 37 | ? | MNHN |
| *Pycnodus apodus* | Bol 94/95 | 26 | 11 | 8 | 62 | 52 | ? | ? | 59 | 45 | 43 | 2 | MNHN |
| *Pycnodus platessus* | BSPG AS I 1208 | 24 | 9 | 8 | 53 | 42 | 40 | 4 | 56 | 44 | 42 | 2 | BSPG |
| *Pycnodus platessus* | BSPG AS I 1209 | 26 | 12 | 8 | 60 | 47 | ? | ? | 58 | 48 | 22 | 2 | BSPG |
| *Pycnodus platessus* | CM 4479 | ? | 12 | 8 | ? | ? | ? | 5 | ? | ? | ? | ? | CM |
| *Pycnodus platessus* | CM 4479a | ? | 12 | 8 | ? | ? | ? | ? | 52 | 41 | ? | ? | CM |
| *Pycnodus gibbus* | CM 4480 | 24 | ? | 8 | 60 | 49 | 45 | 4 | 61 | 50 | 34 | 2 | CM |
| *Pycnodus gibbus* | CM 4480.1 | 26 | 11 | 7 | 59 | 48 | ? | ? | 60 | 48 | 39 | 2 | CM |
| *Pycnodus gibbus* | CM 4481 | 24 | 11 | 8 | 59 | 46 | 35 | 4 | 58 | 46 | 40 | 2 | CM |

| Species name on museum label | Specimen no. | No. of vertebrae | Rib pairs | No. of scale bars | Dorsal fin rays | Anal fin rays | Pectoral fin rays | Pelvic fin rays | Dorsal fin pterygiophores | Anal fin pterygiophores | Caudal fin rays | Arcocentra interdigitations | Museum |
|---|---|---|---|---|---|---|---|---|---|---|---|---|---|
| *Pycnodus* sp. | Coll Baja Pesciara 4 (T.998) | 25 | 13 | 8 | 56 | 44 | 28 | ? | 56 | 43 | 30 | 2 | MCSNV |
| *Pycnodus* sp. | Coll Baja Pesciara 5 (T.999) | 23 | ? | 9 | 55 | 43 | 25 | ? | 58 | 41 | 24 | 2 | MCSNV |
| *Pycnodus* sp. | I.G.135608 | 23 | 9 | 8 | 58 | 46 | ? | 4 | 59 | 58 | 31 | 2 | MCSNV |
| *Pycnodus* sp. | I.G.135609 | 23 | 10 | 10 | 59 | 44 | 24 | 5 | 59 | 41 | ? | 2 | MCSNV |
| *Pycnodus* sp. | I.G.135664 | 26 | 12 | 8 | 49 | 37 | ? | ? | 46 | ? | 30 | ? | MCSNV |
| *Pycnodus* sp. | II D 167 | 27 | 11 | 8 | 51 | 47 | 33 | ? | 51 | 46 | 25 | 2 | MCSNV |
| *Pycnodus* sp. | II D 168 | 25 | ? | 9 | 54 | 44 | ? | ? | 55 | 40 | 25 | 2 | MCSNV |
| *Pycnodus* sp. | II D 170 | 27 | ? | 7 | 59 | 51 | ? | ? | 60 | 47 | 28 | 2 | MCSNV |
| *Pycnodus* sp. | II D 171 | 27 | 11 | 8 | 56 | 42 | ? | ? | 53 | 41 | 24 | 2 | MCSNV |
| *Pycnodus* sp. | II D 180 | 25 | 11 | 9 | 60 | 49 | 32 | 4 | 62 | 50 | 33 | ? | MCSNV |
| *Pycnodus gibbus* | PF 3234 | 25 | 13 | 10 | 54 | ? | 38 | 5 | 56 | ? | 25 | 2 | FMNH |
| *Pycnodus* sp. | (I.G.23??) | 25 | 11 | 9 | 54 | 43 | ? | 4 | 55 | 42 | 23 | ? | MCSNV |
| *Pycnodus* sp. | (I.G.186666) | 26 | 10 | 10 | 46 | 39 | ? | ? | 50 | 42 | 23 | 2 | MCSNV |
| *Pycnodus* sp. | (I.G.186667) | 25 | 11 | 10 | ? | ? | ? | ? | 43 | ? | 27 | 2 | MCSNV |
| *Pycnodus* sp. | (I.G.24497) | 24 | 11 | 9 | ? | ? | ? | ? | 38 | ? | 22 | ? | MCSNV |
| *Pycnodus* sp. | Unknown | 23 | 10 | 8 | 54 | 41 | ? | ? | 51 | 40 | 30 | ? | MCSNV |
| *Pycnodus* sp. | (I.G.135680) | ? | 9 | 10 | ? | ? | ? | ? | ? | ? | ? | ? | MCSNV |
| *Pycnodus* sp. | I.G.37581 | ? | 12 | ? | ? | ? | ? | ? | ? | ? | 23 | ? | MCSNV |
| *Pycnodus* sp. | T.309 | 24 | 11 | 8 | ? | ? | ? | ? | ? | ? | 34 | ? | MCSNV |

**Note:**
Museum abbreviations are mentioned in main text.

one of the last representatives of a basal neopterygian lineage that has been around since at least the Late Triassic (*Tintori, 1981*; *Kriwet, 2001a*; *Poyato-Ariza, 2015*), indicates that pycnodontiforms also produce substantial intraspecific variation similar to living representatives of other ancient actinopterygian lineages such as amiids (*Jain, 1985*) and acipenserids (*Hilton & Bemis, 1999*). Therefore, the identification of different Bolca *Pycnodus* species such as *P. gibbus* (*Heckel, 1856*), may be the result of species over-splitting and can be on the contrary explained by intraspecific variation in meristic counts and ontogeny.

## Habitat use during ontogeny

Our morphometric results show that the morphology of the smaller individuals differ significantly from that of the adults and that *Pycnodus*, like extant actinopterygians, would go through morphological changes throughout ontogeny. Large eye size found in the smaller *Pycnodus* specimens is usually a sign of the specimen being in a juvenile stage as can be seen in many extant teleosts (*Pankhurst & Montgomery, 1990*). Large eye size in pycnodonts has been related to behavioral flexibility and possible nocturnal behavior (*Goatley, Bellwood & Bellwood, 2010*). This could also apply for the Bolca *Pycnodus* although the individuals with the largest eyes (juveniles) are not believed to be more nocturnal as larger eye size in smaller fishes is a natural consequence of ontogeny. The deep body shape of the smaller *Pycnodus* specimens can be interpreted as a sign that the juveniles live within the branches of corals and as they get bigger they start to occupy the water column above the reef. Coral reefs composed of scleractinian coral colonies have been reported in situ (*Vescogni et al., 2016*) and were probably even more extensive based on abundant remains from the laminated and massive fossiliferous limestone from Pesciara and Monte Postale sites. This change to a benthopelagic lifestyle is also supported by the more fusiform body and the narrower caudal peduncle (*Webb, 1982*) seen in larger specimens.

Ecologically similar extant analogues to *Pycnodus*, fishes of the genus *Lethrinus* undergo ontogenetic changes in head shape as they grow in size but their body depth in relation to length does not change drastically during growth (*Wilson, 1998*). The sparid species *Diplodus sargus* and *D. puntazzo* also spend their time as juveniles in crevices in the rocks in shallow water 0–2 m deep and move to rocky bottoms and sea grass beds when adult (*Macpherson, 1998*). However, their ontogenetic trajectory differs from *Pycnodus* as they are more elongate as juveniles and body depth increases with age. Juvenile carangids also have a deeper body than that seen in adults (*Leis et al., 2006*) and are found within lagoonal patch reefs (*Wetherbee et al., 2004*) only moving out of this habitat when larger than 40 cm and becoming more pelagic in their habitat preferences (*Kuiter, 1993*; *Myers, 1999*). Eurasian perch (*Perca fluviatilis*) go through three different feeding modes during their life span; zooplanktivory, benthic macroinvertebrate feeding, and piscivory. The middle stage, benthic feeding results in them shifting to the littoral zone where they have a deeper body and longer fins which aid in maneuverability whereas piscivores and zooplanktivores have a similar body type due to both life stages living in the pelagic realm (*Hjelm, Persson & Christensen, 2000*).

Ontogenetically-related habitat changes also occur in other coral fishes, such as labrids, in which the pectoral fins increase their aspect ratio as these fishes grow in size, enabling them to increase their use of the water column while juveniles stay closer to the bottom (*Fulton, Bellwood & Wainwright, 2001*). Since both juveniles and adults of *Pycnodus* are found in the Bolca Lagerstätte, we hypothesize that unlike many modern coral reef fishes, which significantly change the habitat during ontogeny (*Nagelkerken et al., 2002*; *Dorenbosch et al., 2005a*, *2005b*; *Adams et al., 2006*; *Nagelkerken, 2007*; *Nakamura et al., 2008*; *Shibuno et al., 2008*; *Kimirei et al., 2011*), there is a shift instead in microhabitat use within the reef, in this case juveniles living within coral crevices to adults roaming over the coral reefs.

## CONCLUSION

The quantitative approach here performed supports the hypothesis of *Blot (1987)* that the various *P. nominal* species (*P. apodus*, *P. platessus*, *P. gibbus*) from the Eocene Bolca Konservat–Lagerstätte actually belong to a single species. Due to the holotype of *Pycnodus* being given the specific name of *apoda*, all known specimens of *Pycnodus* from Bolca should be referred to as *P. apodus*. Most of the morphological variation can be explained by the close correlation between morphometric changes and ontogeny, with juveniles and adults occupying different parts of the morphospace. The morphometric differences between juveniles and adults may be due to occupation of different habitats with juveniles sheltering among cover and adults being better adapted to a roaming lifestyle swimming over the benthos to feed. The complex taxonomic history shows that most species typically referred to as *Pycnodus* are different taxa altogether (e.g. all Jurassic and Cretaceous *Pycnodus* specimens) and with the majority of Palaeogene *Pycnodus* being represented by isolated dentition it seems that the only definitive articulated skeletal remains attributed to the genus *Pycnodus* are *P. apodus* from the Bolca Lagerstätte and *Pycnodus* sp. from south-eastern Mexico (*Alvarado-Ortega et al., 2015*). Future studies should analyze other problematic pycnodontiform taxa such as the widely distributed *Gyrodus* from the Middle Jurassic to the Early Cretaceous (*Kriwet & Schmitz, 2005*) to investigate if intraspecific variation might partially explain the supposed diversity of species this genus contains.

## ACKNOWLEDGEMENTS

We would like to thank M. Cerato (BM), Z. Johanson and E. Bernard (NHML), O. Rauhut (BSPG), A. Henrici (CM), L. Grande and W. Simpson (FMNH), M. Fornasiero (MGP-PD), A. Pradel and G. Clément (MNHN), A. Vaccari and R. Zorzin (MCSNV), and U. Göhlich (NHMW) for access to specimens and support while studying these specimens at the museum. We also are grateful to the helpful comments by the reviewers for improving the standard of this paper.

### Funding

This study was supported by a grant of the Austrian Science Fund (FWF): P29796-B29 to Jürgen Kriwet and M2368-B25 to Giuseppe Marramà, SYNTHESYS: FR-TAF-6568 to

John Joseph Cawley. The funders had no role in study design, data collection and analysis, decision to publish, or preparation of the manuscript.

## Grant Disclosures
The following grant information was disclosed by the authors:
Austrian Science Fund (FWF): P29796-B29 and M2368-B25.
SYNTHESYS: FR-TAF-6568.

## Competing Interests
The authors declare that they have no competing interests.

## Author Contributions
- John Joseph Cawley conceived and designed the experiments, performed the experiments, analyzed the data, prepared figures and/or tables.
- Giuseppe Marramà conceived and designed the experiments, performed the experiments, analyzed the data, contributed reagents/materials/analysis tools, authored or reviewed drafts of the paper, approved the final draft.
- Giorgio Carnevale authored or reviewed drafts of the paper, approved the final draft.
- Jürgen Kriwet conceived and designed the experiments, contributed reagents/materials/ analysis tools, authored or reviewed drafts of the paper, approved the final draft.

## Data Availability
The morphometric and landmark data for all the Pycnodus specimens studied are provided in the Supplemental Files.

## Supplemental Information
Supplemental information for this article can be found online at http://dx.doi.org/ 10.7717/peerj.4809#supplemental-information.

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
