# Peer review of "A quantitative approach to determine the taxonomic identity and ontogeny of the pycnodontiform fish Pycnodus (Neopterygii, Actinopterygii) from the Eocene of Bolca Lagerstätte, Italy"

_PeerJ, doi:10.7717/peerj.4809_

## Round 0.1 · original submission · Minor Revisions

Congratulations on putting this interesting dataset and study together. I would like to see this published, but there are crucial aspects, I would like you to address:

Definitions: Please be consistent with the definitions and use of terminology (see also comments by Reviewer 3). You seem to mix up several concepts. For example, ontogenetic variation during growth is something else then intraspecific variation at a single ontogenetic stage. Even if in fossil vertebrates, where each individual specimen can only be measured at a single stage, each individual could still have through ontogeny during lifetime.

Continuous/unimodal distribution: you state that distributions are “continuous” and unimodal (Gaussian). You histograms do not look perfectly gaussian. One can argue if a strictly normal distribution is to be expected – if evidence for a unimodal distribution could not be sufficient. However, if you make such statement, you would at least not do adequately document this and test for it with normality tests. Histograms are not sufficient to evaluate a unimodal distribution(De Baets et al. 2015). In some cases, they can even mask a bimodal distribution (De Baets et al. 2015). Furthermore, part of non-unimodal distribution at certain ontogenetic stages could also relate to sampling – at least in smaller samples, a non-unimodal distribution could give the impression of multimodalit (De Baets,Klug & Monnet 2013). Kernel density estimators might be better for several reasons (Sanvicente-Añorve,Salgado-Ugarte & Castillo-Rivera 2003, Salgado-Ugarte et al. 2000). Beanplot or violin plots even combine kernel density estimators with a boxplot making it possible to evaluate outliers. Note that an approximate Gaussian distribution could also be obtained when lumping different ontogenetic stages or differently aged fishes with similar or slightly offset traits distribution. It might help, to only look at individuals you assume to be adult/non-juvenile or alternatively, only at a larger logarithmic size class, to see if they retain a unimodal distribution. This will make it possible to evaluate how younger specimens might affect the results.

Evaluation of intraspecific variation: Even if all distributions are unimodal, cryptic species could still be buried in your data. Persuing cryptic species might be pointless in many paleontological samples, if we cannot separate them morphologically – not even with particular traits – we will just not pick them up so focusing on units which we can actually separate makes the most sense. Meristic characters could potentially help to disentangle the differences (see also comments by reviewers 1 and 3). This would even be more crucial as you are doing a “cross-sectional” study (Klingenberg 1998) in which each individual specimen is measured at a single stage – opposed to mollusks where you can measure each individual multiple times during growth. This means you even have less data at particular ontogenetic stages. Nevertheless, you might expect that intraspecific variation is unimodal at all ontogenetic stages unless dimorphism or polymorphism at later ontogenetic stages is the case. Nevertheless, in cases of polymorphism, paleontologist might have actually erected two species unless there is good support for it.

Figure size/resolution : Please make sure all your figure are of sufficient size and resolution, particularly figure 2 and 3 are hard to read.

Coral reef interpretation: you interpret microhabitat changes throughout the life. It would just make sure these are consistent with model of reef cover during the time (see comments by reviewer 1). Monte Bolca fishes have been interpreted as earliest reef fishes since long time, but only recently sufficient corals been found “coralgal” buildups (Vescogni et al. 2016) – although these might differ from extant reefs.
Structure: I feel it would be better to place the discussion on Pycnodus after the introduction (see also Reviewer 3).

Cited references: in one case, you have 20 references to make the point that differences in meristic counts are suggestive of intraspecific variation in actinopterygians (line 230-235). In principle, I have no problem with it, but It would make sense to group them according taxonomic groups and stratigraphic age. This would make it more easy to read and more informative.

Please also address the comments in the annotated manuscript, in addition to the reviewer suggestions.

Suggested references:

De Baets K, Bert D, Hoffmann R, Monnet C, Yacobucci MM, Klug C (2015) Ammonoid intraspecific variability. In Klug C, Korn D, De Baets K, Kruta I,Mapes RH (eds) Ammonoid Paleobiology: From anatomy to ecology. Topics in Geobiology 43. Springer, Dordrecht, pp. 359-426
De Baets K, Klug C, Monnet C (2013) Intraspecific variability through ontogeny in early ammonoids. Paleobiology 39:75-94
Klingenberg CP (1998) Heterochrony and allometry: the analysis of evolutionary change in ontogeny. Biological Reviews 73:79-123
Salgado-Ugarte IH, Shimizu M, Taniuchi T, Matsushita K (2000) Size Frequency Analysis by averaged shifted histograms and kernel density estimators. Asian Fish Sci 13:1-12
Sanvicente-Añorve L, Salgado-Ugarte IH, Castillo-Rivera M (2003) The use of kernel density estimators to analyze length-frequency distributions of fish larvae. In Browman HI,Skiftesvik AB (eds) The Big Fish Bang. Proceedings of the 26th Annual Larval Fish Conference. Institute of Marine Research, Bergen, pp. 419-430
Vescogni A, Bosellini FR, Papazzoni CA, Giusberti L, Roghi G, Fornaciari E, Dominici S, Zorzin R (2016) Coralgal buildups associated with the Bolca Fossil-Lagerstätten: new evidence from the Ypresian of Monte Postale (NE Italy). Facies 62:21

.
·

Basic reporting

Sound

Experimental design

Sound

Validity of the findings

Sound

Additional comments

This paper combines a thorough and careful evaluation of the morphometrics of pycnodonts from Bolca with a review of the fossil record of Pycnodus. The subject matter, approach and conclusions are all appropriate. The text is well written and I have no substantive problems with the paper. I do however have a number of suggestions.
L=line
L33 I would break up the two long sentences to improve clarity. The first is a bit repetitive, the second could perhaps be three sentences.
L68 Here and elsewhere there are too many references. Three will suffice. Indeed this point has already been included in numerous reviews.
L87 variation is ?
L98 This paper seeks to determine if the nominal species are morphologically distinct. Surely the first step would be to evaluate the characters used to distinguish the species. Then consider them in a cladistic framework to see of any of the proposed characters and their respective states can reasonably be considered synapomorphies (or autapomorphies depending on the level). Finally, one can consider the overall shape and ontogeny as a possible explanation for observed variation. I cannot see how you can look just at shape and conclude that these are not different species, especially if you predominantly measure characters that were not originally identified as being diagnostic. You may have included all of these diagnostic features (I suspect you did), but it would help to explicitly indicate that you critically examined all features that have been used to separate these nominal species. Currently you have a phenetic evaluation of the shape of individuals. This is not a strong basis for rejecting the various species. They may share 99% of the shape features. All it takes is one small synapomorphic feature and you have a solid rational justification for recognizing a species. I would recommend looking at steps 1,2 and 3 above to provide a more solid justification for your conclusions. NB I think your findings are correct – it is just that this 1,2,3 process would make the final conclusion more water tight.
L201 ‘corroborating definitively’ I suggest you ‘strongly support’
L208 again ‘support’ rather than ‘confirm’
L222 – excessive references
L230 – excessive references
L247 You reject the interpretation of Goatley et al. However, we merely said that “large eyes may suggest that some pycnodont species were nocturnal”. Your data and analyses do not refute this statement. Furthermore, in suggesting that we only looked at juveniles you misinterpret our analyses. We compared eye size residuals in pycnodonts compared to a wide range of actinopterygians. Thus we demonstrated that the eyes of the multiple pycnodont species, and genera, that we examined were relatively large compared to other actinopterygians – given their size. Our analyses specifically allowed for the ontogenetic variation that you are describing. It is axiomatic that smaller fishes have larger eyes. Your data does not offer a basis for rejecting the rather vague conclusions we drew from our data.
It may be better to say that: large eyes in pycnodonts have been related to behavioural flexibility and the possibility of nocturnal behavior (Goatley). This may also apply to the Bolca Pycnodus although the individuals with the largest eyes (juveniles) are not believed to be more likely to be nocturnal as this is a natural consequence of ontogeny; small fishes have larger eyes.
L250 I suspect you overemphasize the extent of coral growth in the Bolca region. In my review of the evolution of coral reefs (Bellwood et al 2017 Biol Rev) I emphasized the low complexity nature of Eocene reefs. I am not certain that there were enough corals for fish to swim between the cracks. In the past we did a study of the early life history of lethrinids (GG Wilson 1998 Rec Aust. Mus.). They predominantly lived in seagrass beds feeding diurnally on the benthos, as adults they rove over the sand around the margins of the reef (feeding both diurnally and nocturnally). I see lethrinids as the ecological (if not morphological) analogs of pycnodonts. So I suspect the small pycnodontids were as you say in a different location – for my money most likely in the widespread seagrass beds; while adults were roving more widely. The in corals vs out of corals scenario has all sorts of problems for me (availability of corals, maneuverability, body shape, feeding mode etc). NB sparids have similar dentition and may be analogs of pycnodontids in the Mediterranean (or in the Caribbean) but they do not play a comparable role in Indo-Pacific reef systems.
L262 again too many refs
L315 gap missing
L354 dentition
L360 ‘supports’ rather than ‘confirms’
L367 I would say “sheltering among cover and adults being better adapted to a roaming lifestyle swimming over the benthos to feed”
Any questions happy to explain, David Bellwood

·

Basic reporting

Good paper, original and relevant, well structured. No comments

Experimental design

High quality and relevant.

Validity of the findings

Data robust, statistically well supported, conclusions well stated. Good way and example of how to tackle the problem of splitting of species because of new names given to growth stages.

Additional comments

Nice and original paper. Indeed Gyrodus would be a good one to tackle the same way.
typo line 118: "points and were selected" ; should be "points were selected"

typo in two words in the references of Geinitz (line 542): Gesellschaft; Isis

Maybe add two additional papers concerning Pycnodus in India:

P. bicresta is first described in:
Kumar, K., and Loyal, R. S. 1987. Eocene ichthyofauna from the Subathu Formation, northwestern Himalaya, India. J. Palaeontological Society of India, 32:60–84.

Furthermore there are some pycnodont teeth described in:
Prasad, G.V.R. & Sahni, A. (1987)
Coastal-plain microvertebrate assemblage from the terminal Cretaceous of Asifabad, Peninsular India. Journal of the Palaeontological Society of India, 32: 5–19

·

Basic reporting

Intro & background are fine. The literature is up to date, but I did not have time to control if they are all well referenced, if all citations are listed or if all listed references are cited.

The structure is fine. I only suggest moving the section "The taxonomic history of Pycnodus", which is very informative and interesting, thus worth being part of this future paper, but beyond the scope of this work. Therefore, I suggest you move it directly after the introduction.

The figures are all relevant, but too small in the pdf version of the manuscript and of low quality when directly downloaded.

The legend to figure 4 "PLS analysis showing a correlation of morphology with ontogeny" should be corrected. The analysis shows the correlation between the morphometric variation and the size of the specimens, which you interpret as ontogeny. Thus, please replace 'morphology' with 'morphometric variatoin' and 'ontogeny' with 'the size of the specimens' or simply 'size'.

Correct figure 5 according to my comments below to incorporate the complete ranges of variation. Alternatively, clearly explain if you are simplifying this variation using intervals and, if so, provide the values for the intervals. This is especially important in your case because such intervals are apparently hiding the actual variation shown by your data, which does not necessarily follow a normal distribution (see below).

To ensure accessibility, the raw data should be please provided in a standard form. The meristic characters are provided with the Table 5, but I do not have the software PAST and have not been able to open the supplementary file with the morphometric data. Even when PAST is a freeware software, not everybody wants to install it or need it to do statistics.

Experimental design

The experimental design is partially problematic. A general problem with the manuscript concerns equaling morphometrics with morphology and morphotype with species. Morphometrics is just one aspect of morphology and morphometrically cryptic species might be well distinguishable on the bases of qualitative morphological features. Therefore, homogeneity in morphometric traits does not necessarily imply that the authors are dealing with a single species. Consequently, I suggest restricting the research question to the analysis of the morphometric spectrum occupied by the sample, which is what the only morphological aspect represented by the data and analyses.

When nominal species have been defined on the bases of morphometric traits only, it is logically correct to put the nominal taxa in synonymy, but one might still hope to complete a more detailed morphological analysis including qualitative features. In the case of this study, according to the authors, Heckel (1856) erected P. gibbus and distinguished it from P. platessus on the bases of two characters: the relative length of the first caudal vertebral apophyses and the proportion between the body depth and the body length. The second character is included in the analyses presented with the manuscript, but the first of these characters is not discussed and, thus, the alleged invalidity of P. gibbus remains questionable.

The results of the morphometric analyses are reasonable and the accompanying figures show a clear overlapping of the morphospaces of the groups delimited a priori according to the labels referring to the nominal species.

GPA and PCA analyses require complete data sets and are not per se able to deal with missing data. It is necessary to replace the missing values in order to operate and different methods have been proposed for that purposes. In the methods section, you should clearly explain which of these methods is being used in the analyses because this decision might affect the results.

Unfortunately, I have some serious concerns about the analyses of the meristic characters. I hope the authors will satisfactorily answer my inquiries.

Table two is said to include "Measurements as percentage of SL …", but counts are not measurements. Also, which is the meaning of those proportions (density) for taxonomic purposes? For example, unless the number of rays in a certain fin increases during ontogeny, which is usually not the case, the proportion will be different between small juveniles and large adults. Please explain the utility and meaning of these ratios.

I noticed more variation in your meristic data than the one represented in the histograms in your figure 5, so I gave a closer look and fast analysis of your counts in your Table 5. Unfortunately, my results are not as uniform as shown in your histograms and I cannot accept your statement about the normal distribution of these characters, which would be a rather rough approximation for some of them.
For example, the distribution of the number of pectoral and dorsal fin rays could equally be part of incomplete Gaussian or bimodal/trimodal distributions. Rather than any of these, the distribution of your data indicates that a lot of information is missing and more sampling is necessary. It should be possible to run at least some normality test with the PAST, isn't it?

Furthermore, your Table 5 includes enormous ranges of variation (even larger than the ones given in your Table 3):
VER RIB SCL DFR AFR PEC PEL DFP AFP CFR
Max 31 14 12 66 52 47 16 65 58 43
Min 20 6 6 17 7 7 2 29 26 15

Such ranges of variation in these counts would be exceptional! How do you explain this? Could this be due to incomplete preservation (not all fin rays or pterygiophores ossified in juveniles or simply not preserved)? If so, why is it reasonable to use these characters? What do they mean? Alternatively, despite the overlapping of the morphospaces, are these ranges rather suggesting interspecific variation?
Fast and simple correlations between these counts and the standard length of the specimens (calculated with Excel) are not significant.

Even after accepting all of the results presented in the manuscriupt, the final statement of the Biometric analyses "Neither morphometric nor meristic characters are therefore useful in determining two or more different morphologically identifiable morphotypes within Pycnodus, corroborating definitively Blot's (1987) hypothesis that only one species (P. apodus; see also below) is present in the Bolca Lagerstätte" (lines 200-202) is problematic and implies some serious misconceptions. First of all, the normal distribution of each variable (meristic or morphometric) and the positive linear regression of the individual morphometric characters respect to the total or standard lengths do not necessarily imply that the multivariate distribution will be homogeneous. Second, as explained above, the morphometric homogeneity of the data does not necessarily imply the presence of a single species and the morphometric analysis alone is not enough to solve the taxonomic question (qualitative characters have not been analyzed, including a diagnostic feature given by Heckel for P. gibbus).

Please note that even when the statement is wrong, you might be right in your taxonomic conclusions, but you still need to demonstrate that with a more complete and thorough study.

Validity of the findings

The authors carried out morphometric and meristic analyses of numerous specimens referred to three nominal species from the Eocene of Monte Bolca within the genus Pycnodus. This is a very interesting approach to taxonomy and it is certainly significant to decide on the validity of nominal taxa that are defined on the bases of morphometric and/or meristic characters exclusively, but this is not the case of the nominal species included in the analysis. Still, it is valid to propose the potential synonymy of these species, pending a morphological analysis of qualitative traits.

Also, outliers are mentioned, but not discussed. How are those outlying specimens related to the type specimens on which the nominal species are based and how good do they fit the putative diagnostic features of these nominal taxa? A discussion of the outliers is necessary and would represent a valuable contribution to the manuscript.

Despite the criticisms concerning the potentially misleading taxonomic conclusions, the morphometric study is valuable beyond the taxonomic purposes, which is shown by the very interesting conclusions about different ecological preferences between juveniles and adult specimens.

Additional comments

To summarize the main concerns I have with your manuscript, I strongly recommend more caution in your taxonomic conclusions, you should complete the methodological information, please provide the raw morphometric data in a standard form and thoroughly discuss the meaning of the meristic variation represented by the data, which is very high and, contrary to your conclusions, more luckily suggests inter- rather than intraspecific variation. After addressing these main questions, and the other minor revisions I indicated above and directly in the manuscript, I strongly recommend publication of this manuscript, which will then represent a valuable contribution to our knowledge of pycnodont fishes, the Eocene fossil association of the Monte Bolca, and will of course represent a good example for morphometric studies in general.
I'm looking forward for the future studies announced at the end of this manuscript, but please be aware of the conceptual differences behind the terms 'morphology', 'morphometrics', 'morphotype' and 'species'!

---

## Round 0.2 · Minor Revisions

Thank you for addressing our suggestions. They made the manuscript even better than it already was. Your manuscript is as good as accepted - i just wanted you to make some additional changes before publication. Most are just minor formatting issues, but in some cases it would be scientifically appropriate to add references for particular statements (see annotated pdf). In this case, it is not about which references are cited, but that at least one reference relevant to make or support this statement is added.

These include:

Line 250: the use of kernel density estimators should also be mentioned here in the methods and it would be appropriate to add a reference here or later in the text on its use in addition to histograms. I suggested one i was familiar with, but i am sure there a tons of alternatives who could be cited instead:

Salgado-Ugarte, I. H., Shimizu, M., Taniuchi, T., and Matsushita, K., 2000, Size Frequency Analysis by averaged shifted histograms and kernel density estimators: Asian Fisheries Science, v. 13, p. 1-12.

Line 301: i would also be appropriate to cite a reference who discussed influence of sample size on significant high frequency of intermediate values. I suggested one i was involved and therefore familiar with, but there are probably various alternative ones who should have discussed this and could be cited instead:

De Baets, K., Klug, C., and Monnet, C., 2013, Intraspecific variability through ontogeny in early ammonoids: Paleobiology, v. 39, no. 1, p. 75-94.

The most crucial one being your coral reef interpretations (after Line 390: "above the reef"), which forms (and should stay) an important of your interpretations: It might be trivial to you that coral reefs must have been extensive in Bolca during the time-interval you studied, but not to the reader. I have been there regularly on excursions with students and i am familiar with much of the literature, but this aspect is rarely discussed in appropriate detail. As it would further support (or at least a crucial point for) your hypothesis, it is crucial to add a small line on the extent of coral reefs at Bolca to your manuscript with appropriate reference(s) - i really so not valid reason not to include at least reference in support of the extent of coral reefs in Bolca. Vescogni et al. (2016) would be an obvious choice, but if you are aware of alternatives that it is also fine:

Vescogni, A., Bosellini, F. R., Papazzoni, C. A., Giusberti, L., Roghi, G., Fornaciari, E., Dominici, S., and Zorzin, R., 2016, Coralgal buildups associated with the Bolca Fossil-Lagerstätten: new evidence from the Ypresian of Monte Postale (NE Italy): Facies, v. 62, no. 3, p. 21.

Please address these and other changes suggested in the annotated pdf.

---

## Round 0.3 · Minor Revisions

Thank you for taking into account my suggestions. There are still some issues with the raw data. The raw landmark data files are missing - 18 landmarks and 14 semi landmarks were collected for each specimen but there should be a matrix of 32 landmarks for 39 specimens. Also, supplementary table S1 only provides the 19 measurements, not the raw data for the ten meristic counts (Number of vertebrae, ribs, scale bars, paired fin rays, median fin rays, median fin pterygiophores and caudal fin rays). As far as I understand also only mean values have been uploaded for the meristic data. Please provide these as raw data also.

---

## Round 0.4 · accepted · Accept

Thank you for making these final changes and uploading the landmark data for all specimens as supplementary material.

#